# Metabolic Syndrome Prevention Potential of Tamarillo: Phytochemical Composition, Antioxidant Activity, and Enzyme Inhibition Before and After Digestion

**DOI:** 10.3390/foods14071282

**Published:** 2025-04-07

**Authors:** Shin-Yu Chen, Qi-Fang Zhang, Hui-Shan Shen, Sheng-Dun Lin

**Affiliations:** 1Department of Food Science, National Pingtung University of Science and Technology, Pingtung 912301, Taiwan; sychen@mail.npust.edu.tw; 2Department of Food Science and Technology, Hungkuang University, 1018, Sec. 6, Taiwan Boulevard, Shalu District, Taichung 433304, Taiwan; 0712fa520@gmail.com (Q.-F.Z.); roxenshen33@gmail.com (H.-S.S.)

**Keywords:** tamarillo, metabolic syndrome, phytochemicals, enzyme inhibition, gastrointestinal digestion, bioavailability

## Abstract

Tamarillo (*Solanum betaceum* Cav.) is rich in polyphenols, anthocyanins, and carotenoids, making it a promising candidate for functional food development. This study investigated phytochemical profiles and bioactivities in different tamarillo parts. Various parts of tamarillo were extracted using water and ethanol (0–95%), with 95% ethanol yielding the highest content of bioactive compounds in the peel, pulp, mucilage, and whole fruit, while 75% ethanol was more effective for the seeds. Among tamarillo components, the peel exhibited the highest concentrations of hydroxycinnamoyl derivatives, anthocyanins, and carotenoids, along with superior antioxidant capacity, including strong scavenging activity against 2,2-diphenyl-1-picrylhydrazyl (DPPH) radicals (EC_50_, 45.26 µg extract/mL) and high reducing power (EC_50_, 113.3 µg extract/mL). The peel extract exhibited the strongest inhibitory effects on α-glucosidase (IC_50_, 1.623 mg/mL) and angiotensin-converting enzymes (IC_50_, 1.435 mg/mL). In contrast, the pulp extract demonstrated the highest inhibitory activity against pancreatic lipase (IC_50_, 0.882 mg/mL) and α-amylase (IC_50_, 2.369 mg/mL). These findings suggest that tamarillo extracts possess potent antioxidant activity and enzyme-inhibitory properties related to metabolic syndrome (MetS). However, gastrointestinal digestion simulation influenced the bioactive compound content and bioactivities. Overall, tamarillo has promising potential as a functional ingredient for MetS prevention, but processing strategies are needed to retain its bioactive properties.

## 1. Introduction

The modern lifestyle is characterized by a high intake of processed foods, exposure to environmental chemicals, and insufficient physical activity, which has been linked to increased oxidative stress [1]. Oxidative stress (OS) results from excessive free radical production or reduced antioxidant defenses and is often accompanied by chronic inflammation. Together, these factors contribute to the development of hyperlipidemia, hypertension, and impaired glucose tolerance, forming the foundation of metabolic diseases [2,3]. As a result, the prevalence of metabolic syndrome (MetS) has risen sharply in recent decades, posing a significant public health challenge [4]. One-fifth of adults in the United States and Europe are affected, with global prevalence estimates ranging from 12.5% to 31.4% [5]. Alarmingly, in 2020, approximately 3% of children and 5% of adolescents worldwide were also diagnosed with MetS [6], highlighting the increasing burden of this condition at an early age and the urgent need for MetS prevention.

MetS is a major driver of the global cardiovascular disease crisis, substantially increasing the risk of type 2 diabetes, cardiovascular disease, and premature death. It is characterized by interconnected cardiometabolic abnormalities, including hyperglycemia (insulin resistance), dyslipidemia (reduced serum high-density lipoprotein cholesterol and elevated triglycerides), and hypertension, often accompanied by abdominal obesity. The diagnosis of MetS requires the presence of at least three of these metabolic abnormalities [5,6,7].

In Taiwan, according to the Ministry of Health and Welfare, MetS-related diseases accounted for 27.4% of all deaths in 2023, based on the top ten leading causes of mortality. These included heart disease (excluding hypertensive diseases) (11.4%), cerebrovascular disease (6.0%), diabetes (5.7%), and hypertensive diseases (4.3%). Notably, this proportion was comparable to that of malignant neoplasms (25.8%), underscoring the severe public health implications of MetS [8].

Beyond its impact on chronic disease prevalence and mortality, MetS imposes a substantial economic burden. Global healthcare expenditures and the loss of potential economic productivity related to MetS amount to trillions of dollars [4]. Currently, there is no single pharmacological treatment for MetS, and its management typically requires multiple medications. However, polypharmacy may lead to adverse effects and decreased patient adherence to medication [9]. Therefore, developing effective strategies for disease prevention has become a critical research priority. One promising approach is the development of functional foods specifically designed to mitigate the risk of MetS-related diseases.

Numerous studies have widely recognized fruits, vegetables, and other plant-based products as key dietary components for preventing MetS due to their abundance of phytochemicals, also known as natural bioactive compounds. Phytochemicals can be categorized into several classes based on their chemical structure and biological activity, including polyphenols, carotenoids, alkaloids, phytosterols, nitrogen-containing compounds, and organosulfur compounds. These plant-derived compounds exhibit diverse biological activities and provide various health benefits [9,10,11].

OS has been strongly associated with the pathogenesis of numerous diseases, including not only MetS but also cancer, atherosclerosis, malaria, Alzheimer’s disease, rheumatoid arthritis, neurodegenerative disorders, and preeclampsia [12]. Antioxidants help prevent chronic degenerative diseases by counteracting OS. An antioxidant is defined as “a redox-active compound that limits oxidative stress by reacting non-enzymatically with a reactive oxidant” [3]. Phytochemicals act as potent antioxidants and may help mitigate both inflammation and OS [13]. Therefore, these compounds have demonstrated protective effects against various diseases, including obesity, diabetes, renal disorders, and cardiovascular disease [9,10]. Consequently, recent research has increasingly emphasized the important role of phytochemicals in mitigating MetS progression, highlighting their long-term beneficial effects in managing this condition [9,10,11]. Among phytochemicals, polyphenols have been the most extensively studied [9]. They have been reported to play a significant role in preventing the development of MetS by lowering plasma glucose levels, reducing blood pressure and body weight, and improving dyslipidemia [13,14,15]. Additionally, carotenoids have been suggested as important supportive compounds in treating MetS and other metabolic disorders [16].

Tamarillo, also known as the tree tomato (*Solanum betaceum* Cav. or *Cyphomandra betacea* Sendt.) [17,18], is native to South America and is cultivated in several other regions worldwide for its nutrient-rich fruit [19]. Tamarillo is a rich source of fiber, vitamin C, potassium, phenolic compounds, anthocyanins, carotenoids, and other phytochemicals while also being low in fat and sodium [19,20,21,22]. Moreover, it has a relatively high polyphenol content than other tropical fruits [19]. Due to its rich nutritional and bioactive composition, tamarillo has been proven to exhibit high antioxidant activity [20], suggesting its potential as a natural antioxidant with protective effects against metabolic syndrome [22] and cancer [23]. Additionally, different parts of the tamarillo plant exhibit variations in phytochemical composition and content [22]. Furthermore, tamarillo wastes and by-products have also been reported to contain a rich nutritional and bioactive composition [21]. This indicates that tamarillo is abundant in phytochemicals, offering health benefits and potential protection against chronic diseases.

The health benefits of these phytochemicals depend on their purity. Therefore, they have significant applications after extraction, developing functional foods and nutraceuticals [24]. Moreover, the functionality of these bioactive compounds is influenced not only by their concentrations but also by their bioaccessibility and bioavailability after gastrointestinal digestion [25]. Understanding the factors that influence the bioavailability of phytochemicals is crucial for evaluating their biological significance and efficacy as functional food ingredients. Therefore, this study aimed to investigate the effects of varying ethanol concentrations in the extraction solvents and simulated gastrointestinal digestion in the phytochemical composition and bioactivity of different parts of tamarillo. First, the most suitable water-to-ethanol ratio for extracting the highest amount of phytochemicals was determined, followed by an assessment of their antioxidant capacity and inhibitory effects on key enzymes associated with MetS, including pancreatic lipase (hyperlipidemia), α-amylase and α-glucosidase (hyperglycemia), and angiotensin-converting enzymes (ACEs) (hypertension). Finally, a simulated gastrointestinal digestion model was employed to evaluate the impact of digestion on the phytochemical content in different parts of tamarillo and to determine whether digestion influences their enzyme-inhibitory activity related to MetS.

## 2. Materials and Methods

### 2.1. Materials

Farmers in Nantou, Taiwan, donated red tamarillos (*S. betaceum* Cav.) harvested 22–24 weeks post-anthesis at an altitude of 1100–1200 m above sea level. The fruit trees were five years old. Tamarillo fruits were stored at 4 °C to 6 °C, and their components were separated within a week. After being washed and drained, the fruits had their stems removed before being cut in half using a knife. The mucilage and seeds in each fruit’s center were scooped out with a small spoon, and the pulp was collected. The mucilage and seed mixture were then separated using fine mesh laundry bags. Appendix A presents the images of various parts of fresh tamarillo and their freeze-dried powders, reproduced from our previously published study [22].

### 2.2. Chemicals

Ethanol and methanol were purchased from Avantor Performance Materials (Radnor, PA, USA). Acetone, acarbose, α-amylase (from porcine pancreas, Type VI-B), α-carotene, α-glucosidase (from *Saccharomyces cerevisiae*), α-tocopherol, ascorbic acid, β-carotene, β-cryptoxanthin, butylated hydroxyanisole (BHA), captopril, chlorogenic acid, cyanidin-3-O-rutinoside, delphinidin-3-O-rutinoside, dimethyl sulfoxide (DMSO), 2,2-diphenyl-1-picrylhydrazyl (DPPH), Folin–Ciocalteu phenol reagent, gallic acid, hippuric acid, hippuryl-L-histidyl-L-leucine (HHL), iron (III) chloride, lipase (from porcine pancreas, Type II), lycopene, orlistat, *p*-nitrophenyl-α-D-glucopyranoside (*p*-NPG), pancreatin (from porcine pancreas, 4 × USP specifications), pelargonidin-3-O-rutinoside, pepsin (from porcine gastric mucosa, 4711 U/mg protein), potassium ferricyanide, porcine bile extract, sodium chloride, starch (from potato), 3,5-dinitrosalicylic acid (DNS), trichloroacetic acid (TCA), Triton X-100, and ACE (from rabbit lungs) were purchased from Sigma-Aldrich (St. Louis, MO, USA). Rosmarinic acid and zeaxanthin were purchased from ChromaDex, Inc. (Irvine, CA, USA), and β-cryptoxanthin was obtained from Extrasynthese Co. (Genay Cedex, France). Sodium carbonate, sodium dihydrogen phosphate, and sodium phosphate were obtained from Union Chemical Works Ltd. (Hsinchu City, Taiwan). Ethyl acetate was purchased from Macron Fine Chemicals™ (Center Valley, PA, USA), and n-hexane was obtained from Tedia Co., Inc. (Fairfield, OH, USA). Hydrochloric acid was purchased from Showa Chemical Co., Ltd. (Tokyo, Japan). Dialysis bags (molecular weight cut-off of 10,000 Da) were obtained from Membrane Filtration Products, Inc. (Seguin, TX, USA).

### 2.3. Sample Preparation and Extraction

The samples were subjected to vacuum freeze-drying using a freeze-dryer (FD-20L-6S, Kingmech Co., Ltd., New Taipei, Taiwan) under the following conditions: a vacuum pressure < 0.2 mmHg, a chamber temperature of 25 °C, and a condenser temperature of −50 °C. Dried samples were ground using a mill (RT-30HS, Rong Tsong Precision Technology Co., Taichung, Taiwan) and passed through a 0.7 mm sieve. The resulting powders were sealed in PET/Al/PE laminated bags and stored at −25 °C until further extraction.

The dried samples were extracted using water and ethanol at different concentrations (25%, 50%, 75%, and 95%) at a 1:10 (*w*/*v*) ratio in a 75 °C water bath shaker (SB302, Kansin Instruments, Kaohsiung, Taiwan) at 150 rpm for 30 min. After centrifugation (8000 rpm, 10 min; 8060× *g*) using a centrifuge (Himac CR21G, Hitachi High-Technologies Co., Ltd., Minato-Ku, Tokyo, Japan) and filtration, the remaining residues were re-extracted with the same solvent under identical conditions. The supernatants from both extractions were pooled, concentrated under reduced pressure, and subjected to vacuum freeze-drying using a freeze dryer (N-1000, Rikakikai Co., Ltd., Tokyo, Japan). The resulting freeze-dried tamarillo extracts were designated 0E, 25E, 50E, 75E, and 95E, corresponding to extractions performed with 0%, 25%, 50%, 75%, and 95% ethanol, respectively. In addition, the peel, pulp, mucilage, seeds, and whole fruit were abbreviated as P, L, M, S, and W, respectively.

### 2.4. Determination of Phytochemicals in Tamarillo Extracts

#### 2.4.1. Total Phenols in Tamarillo Extracts

The sample solution preparation was modified based on Vasco et al.’s method (2009) [26]. Precisely weighed amounts of freeze-dried tamarillo extracts were used—0.02 g of peel, 0.2 g of pulp and mucilage, 0.06 g of seeds, and 0.1 g of whole fruit. Each sample was mixed with 4 mL of 50% methanol and subjected to ultrasonic shaking for 15 min, followed by centrifugation (4000 rpm, 25 °C; 2600× *g*) for 15 min. The supernatant was collected, while the residue was re-extracted with 70% acetone under the same conditions. The two supernatants were combined and diluted to a final volume of 10 mL with deionized water.

The determination of total phenols was performed according to the method of Mau et al. (2017) [27]. Briefly, 200 μL of either standard or sample solution was mixed with 400 μL of Folin–Ciocalteu reagent (2 N) and allowed to stand for 1 min. Subsequently, 6 mL of 5% Na_2_CO_3_ solution was added, and the mixture was left to stand for 60 min. The absorbance was then measured at 760 nm, and the total phenolic content was calculated based on a calibration curve constructed using gallic acid as the standard.

#### 2.4.2. Total Anthocyanins in Tamarillo Extracts

The sample solution preparation was modified using Shao et al.’s method (2014) [28]. A precisely weighed 0.3 g of freeze-dried tamarillo extract was placed in a test tube and extracted with 3 mL of methanol containing 1 M HCl using ultrasonic shaking for 15 min. The mixture was centrifuged at 5372 rpm (4000× *g*, 25 °C) for 15 min. The supernatant was transferred to a separate test tube, and the extraction was repeated four additional times. The five supernatants were combined and diluted to a final volume of 20 mL with methanol containing 1 M HCl for analysis.

Total anthocyanin content was determined using the pH differential method described by Mau et al. (2017) [27], with slight modifications. Briefly, 2 mL of each extract (from different fruit parts and whole fruit) was diluted to 10 mL with pH 1.0 and pH 4.5 buffer solutions, respectively. The absorbance of each solution was measured at 520 nm and 700 nm. Total anthocyanin content was expressed as cyanidin-3-O-glucoside equivalent and calculated using the following formula:Monomeric anthocyanin pigment (mg/g) = [A × MW × DF × 1000/(ε × *l*)] × (V/sample weight)
where

A = (A_520nm_ − A_700nm_)_pH1.0_ − (A_520nm_ − A_700nm_)_pH4.5_

MW is the molecular weight of cyanidin-3-O-glucoside (449.2 g/mol).

DF is the dilution factor.

ε is the molar extinction coefficient for cyanidin-3-O-glucoside (26,900 L/(cm × mol)).

*l* is the path length (1 cm).

V is the final volume of extract (L).

1000 is the conversion factor to express results in mg/g.

#### 2.4.3. Total Carotenoids in Tamarillo Extracts

The preparation and determination of total carotenoids were modified from the method of Knockaert et al. (2012) [29]. An amount of 0.05 g of freeze-dried tamarillo extract was precisely weighed from each part, and the whole fruit was mixed with 4 mL of hexane:acetone:ethanol (2:1:1, *v*/*v*) and 1 mL of 10% KOH. The mixture was subjected to ultrasonic shaking for 15 min, followed by centrifugation (5000 rpm, 4 °C; 3466× *g*) for 5 min. The supernatant was transferred to a test tube, and the extraction was repeated four additional times. The five supernatants were combined and diluted to a final volume of 25 mL with the extraction solvent for analysis.

The determination of the total carotenoids was performed according to the method of Knockaert et al. (2012) [29], with slight modifications. Briefly, 25 mL of the carotenoid extract from each processing stage was transferred into a separatory funnel, followed by the addition of 25 mL of deionized water. The mixture was vigorously shaken and then allowed to stand for 30 min. The upper organic phase was collected, and the absorbance was measured at 450 nm. Total carotenoid content was expressed as β-carotene equivalent (βCE) and calculated using the following formula:$$\mathrm{Carotenoid}\;\mathrm{concentration}\;(\mathrm{mg}/g)=[(A\;\times \;V\;\times \;10^{4})/(E_{1cm}^{1\%}\times \text{sample}\;\text{weight})]/1000$$
where:

A is the absorbance at 450 nm.

V is the volume of the extract (mL).

$E_{1cm}^{1\%}$ is the extinction coefficient for β-carotene in hexane (2560).

#### 2.4.4. Individual Phytochemicals

##### Hydroxycinnamoyl Derivatives Determination

The sample preparation and HPLC analysis was conducted based on the method of Espín et al. (2016) [30] and Chen et al. (2024) [22], with slight modifications. An accurately weighed 0.1 g of freeze-dried tamarillo extract was placed in a test tube and mixed with 2 mL of 100% methanol. The mixture was sonicated for 15 min, then brought to a final volume of 5 mL with 100% methanol. After centrifugation at 3000 rpm (1400× *g*) for 10 min at 25 °C, the supernatant was filtered through a 0.45 μm syringe filter and subjected to HPLC–DAD analysis. The system was equipped with a Hitachi L-2130 pump and a Hitachi L-2455 diode array detector, along with a Luna 5 µm C18(2) column (250 mm × 4.6 mm, 5 µm particle size; Phenomenex, Torrance, CA, USA). The column temperature was maintained at 35 °C. The mobile phase consisted of 0.1% formic acid in water (solvent A) and acetonitrile (solvent B), delivered at a flow rate of 0.5 mL/min using the following gradient: 15% B (0–5 min), 20% B (10 min), 35% B (20 min), 50% B (30 min), 60% B (35–40 min), and re-equilibration to 15% B (50 min). Detection was carried out at 330 nm, and the injection volume was 10 μL, using a Hitachi L-2200 autosampler. Compounds were identified by comparing retention times and UV spectra with those of authentic standards. Quantification was based on external calibration curves prepared for each compound.

##### Anthocyanin Determination

The sample solution preparation and the chromatographic procedure were modified using the method proposed by Espín et al. (2016) [30] and Chiou et al. (2018) [31]. Anthocyanins were extracted by weighing 0.1 g of freeze-dried tamarillo extract and adding 2 mL of methanol and 0.1% formic acid mixture (75:25, *v*/*v*), followed by sonication for 15 min. The extract was then adjusted to a final volume of 5 mL with the same solvent mixture and centrifuged at 1400× *g* for 10 min at 25 °C, and the supernatant was filtered through a 0.45 μm syringe filter prior to HPLC analysis. Analysis was performed using a Hitachi L-2130 pump and L-2455 diode array detector equipped with a Lichrospher^®^ 100 RP-18e column (250 mm × 4 mm, 5 μm; Merck, Darmstadt, Germany), maintained at 30 °C. The mobile phase consisted of 10% formic acid (solvent A) and methanol (solvent B), with a flow rate of 1.0 mL/min. Gradient elution was programmed as follows: 5% B at 0 min, increased to 60% B at 20 min, 100% B from 25 to 30 min, and returned to 5% B from 35 to 40 min. Detection was carried out at 520 nm. Anthocyanins were identified by comparing their retention times and UV spectra with those of authentic standards and were quantified using external calibration curves.

##### Carotenoid Determination

The chromatographic procedure was based on the method of Lin and Chen (2005) [32] and Chen et al. (2024) [22], with slight modifications. Carotenoids were extracted by accurately weighing 0.6 g of freeze-dried tamarillo extract in a test tube, followed by the addition of 2 mL of ethyl acetate. The mixture was sonicated for 15 min and then brought to a final volume of 5 mL with ethyl acetate. The extract was centrifuged at 2600× *g* (4000 rpm) for 10 min at 25 °C. The resulting supernatant was filtered through a 0.45 μm syringe filter, stored under refrigeration, and subjected to HPLC–DAD analysis. Analysis was performed using a Hitachi L-2130 pump and L-2455 diode array detector equipped with a Lichrospher^®^ 100 RP-18e column (250 mm × 4 mm, 5 μm; Merck, Darmstadt, Germany), maintained at 30 °C. The mobile phase consisted of acetonitrile, methanol, and ethyl acetate (75:15:10, *v*/*v*/*v*) delivered at a flow rate of 1.0 mL/min. Detection was carried out at 450 nm. Carotenoids were identified by comparing their retention times and UV spectra with those of authentic standards and quantified based on calibration curves constructed from standard compounds.

### 2.5. Determination of Antioxidant Activity

#### 2.5.1. Determination of Scavenging Ability on DPPH

Freeze-dried tamarillo extracts and standard compounds, including ascorbic acid, BHA, and α-tocopherol, were precisely weighed, dissolved in 100% methanol, and diluted to appropriate concentrations. The scavenging ability of the samples against DPPH-free radicals was analyzed according to Shimada et al.’s method (1992) [33], and both the scavenging activity and the median effective concentration (EC_50_) required to achieve 50% scavenging were calculated. Dose-response experiments were conducted to assess the efficacy of the compounds, with EC_50_ representing the concentration required to elicit 50% of the maximum response. These data were typically analyzed using a linear regression analysis, in which EC_50_ serves as a key parameter describing the compound’s activity [34].

#### 2.5.2. Determination of Reducing Power

Freeze-dried tamarillo extracts and standard compounds, including ascorbic acid, BHA, and α-tocopherol, were precisely weighed, dissolved in 100% methanol, and diluted to appropriate concentrations. The reducing power of the samples was analyzed according to Oyaizu et al.’s method (1986) [35]. The EC_50_ value, defined as the sample concentration required to reach an absorbance of 0.5 at 700 nm, was used as an indicator of reducing power [36].

### 2.6. Inhibitory Effects on Enzyme Activities in Tamarillo Extracts

#### 2.6.1. Pancreatic Lipase

For the pancreatic lipase inhibition assay, P95E, L95E, M95E, and W95E extracts (0.1 g each) and S75E extract (0.4 g) were weighed in glass test tubes and dissolved in 0.75 mL of 5% DMSO. The mixtures were sonicated for 20 min and then diluted to 5 mL with 5% DMSO, followed by centrifugation at 4000 rpm (≈2600× *g*) for 15 min at 25 °C. The resulting supernatants had stock concentrations of 20 mg/mL for P95E, L95E, M95E, and W95E and 80 mg/mL for S75E. These stock solutions were further diluted with 5% DMSO to obtain a range of concentrations for the enzyme assay. Pancreatic lipase inhibitory activity was measured using a modified protocol based on McDougall et al. (2009) [37] and Luyen et al. (2013) [38], with orlistat as the positive control. Percentage inhibition at each sample concentration was recorded, and the half-maximal inhibitory concentration (IC_50_) was calculated.

#### 2.6.2. α-Amylase

For the α-amylase inhibition assay, P95E, L95E, M95E, and W95E extracts (0.12 g each) and the freeze-dried S75E extract (0.4 g) were weighed in test tubes and dissolved in 1 mL of 5% DMSO. The mixtures were sonicated for 15 min, and then diluted to 5 mL with 0.02 M Na_2_HPO_4_ buffer (pH 6.9, containing 0.006 M NaCl) and centrifuged at 4000 rpm (2600× *g*) for 15 min at 25 °C. The resulting supernatants had stock concentrations of 24 mg/mL for P95E, L95E, M95E, and W95E and 80 mg/mL for S75E. These stock solutions were further diluted with the same phosphate buffer to yield the required concentrations for the assay. α-Amylase inhibitory activity was measured using a modified method established by Pinto et al. (2010) [39], with acarbose as the positive control. The percentage inhibition at each concentration was determined, and IC_50_ values were calculated.

#### 2.6.3. α-Glucosidase

For the α-glucosidase inhibition assay, freeze-dried tamarillo extract (0.125 g) was dissolved in 1 mL of 5% DMSO and sonicated for 15 min. The mixture was diluted to 5 mL with 0.1 M phosphate-buffered saline (PBS, pH 6.9) and then centrifuged at 4000 rpm (2600× *g*) for 15 min at 25 °C. The resulting supernatant had a stock concentration of 25 mg/mL, which was further diluted with the same PBS to prepare a range of concentrations for the assay. α-Glucosidase inhibitory activity was measured using a modified method established by Chiang et al. (2014) [40], with acarbose as the positive control. Percentage inhibition was measured at each concentration, and IC_50_ values were calculated.

#### 2.6.4. Angiotensin-Converting Enzyme (ACE)

For the ACE inhibition assay, freeze-dried tamarillo extract (0.09 g) was dissolved in 1 mL of 5% DMSO and sonicated for 15 min. The mixture was diluted to 5 mL with 300 mM NaCl buffer (pH 8.3) and centrifuged at 4000 rpm (2600× *g*) for 15 min at 25 °C. The resulting supernatant had an approximate stock concentration of 18 mg/mL, and it was further diluted with the NaCl buffer to obtain various concentrations for the assay. ACE inhibitory activity was measured using a modified method established by Actis-Goretta et al. (2006) [41], with captopril as the positive control. The percentage inhibition at each concentration was determined, and IC_50_ values were calculated.

### 2.7. Simulation of In Vitro Gastrointestinal Digestion and Analysis of Composition and Bioactivity

#### 2.7.1. In Vitro Gastrointestinal Digestion

The in vitro gastrointestinal digestion process was simulated based on the methods established by Bouayed et al. (2011) [42] and Chiang et al. (2014) [40], with modifications. Tamarillo extracts underwent simulated gastric and intestinal phases, and the resulting digested samples, including the supernatant, insoluble residue, and dialysate fractions, were collected. These samples were analyzed for their bioactive components, including total phenols, total anthocyanins, and total carotenoids, as well as their inhibitory activity against metabolic syndrome-related enzymes.

In the gastric phase, 200 mg of tamarillo extract was mixed with 50 mL of 0.9% NaCl, 4 mL of 0.1 M HCl, and 4 mL of pepsin solution (20 mg/mL in 0.1 M HCl) in serum bottles. The mixture was incubated in a shaking water bath at 37 °C for 1 h (100 rpm), maintaining a pH of 2.0–2.5.

In the intestinal phase, a 15.5 cm dialysis bag (molecular weight cut-off of 10 kDa) was pre-soaked in 0.9% NaCl solution, sealed at one end with a clip, and filled with 5.5 mL of NaCl and 5.5 mL of 0.5 M NaHCO_3_. After removing air bubbles, the dialysis bag was sealed and immediately immersed in the gastric digest. The mixture was incubated in a shaking water bath at 37 °C for 45 min (100 rpm) to simulate the transition from the gastric phase to the intestinal phase, during which the pH was adjusted to approximately 6.5. Subsequently, 18 mL of a pancreatin and porcine bile extract mixture (2 mg/mL pancreatin, 12 mg/mL bile extract dissolved in 0.1 M NaHCO_3_) was added, and digestion continued in the shaking water bath at 37 °C for 2 h (100 rpm), with the final pH maintained at 7.0–7.5. After digestion, the mixture was centrifuged at 10,000× *g* for 20 min to separate the soluble supernatant and the insoluble precipitate. The collected fractions were stored at −20 °C to further analyze bioactive compounds, antioxidant capacity, and enzyme inhibition activity. The fraction diffused in the dialysis bag was considered the serum-available portion, representing compounds that could potentially enter systemic circulation [43].

#### 2.7.2. Determination of Phytochemicals in Digested Extracts

##### Total Phenols in Digested Extracts

The sample preparation and determination followed the method described in Section 2.4.1, with slight modifications in sample weight. The weights of the dialysate samples were 0.07 g for P95E, 0.3 g for L95E and M95E, 0.2 g for S75E, and 0.2 g for W95E. For the intestinal supernatant, the weights were 0.06 g for P95E and 0.4 g for L95E, M95E, S75E, and W95E. The weights of the intestinal precipitate were 0.012–0.015 g for P95E, 0.013 g for L95E, 0.011 g for M95E, 1.00–1.22 g for S75E, and 0.011–0.012 g for W95E.

##### Total Anthocyanins in Digested Extracts

The sample preparation and determination followed the method described in Section 2.4.2, with adjustments to the sample weight. The weights of the dialysate samples were 0.23 g for P95E, 0.3 g for L95E, S75E, and W95E, and 0.273 g for M95E. For the intestinal supernatant, the weight was 0.3 g for freeze-dried digested samples from all tamarillo parts. The weights of the intestinal precipitate were 0.024 g for P95E, 0.067 g for L95E, 0.055 g for M95E, 0.2 g for S75E, and 0.054 g for W95E.

##### Total Carotenoids in Digested Extracts

The sample preparation and determination followed the method described in Section 2.4.3, with modifications to sample weight. The dialysate and intestinal supernatant samples were weighted at 0.1 g for freeze-dried digested material from all tamarillo parts. The weights of the intestinal precipitate were 0.022 g for P95E, 0.05 g for L95E, M95E, and S75E, and 0.043 g for W95E.

#### 2.7.3. Inhibitory Effects on Enzyme Activities in Digested Extracts

The determination followed the method described in Section 2.6.

### 2.8. Statistics

Each measurement was performed in triplicate. The variance in the experimental data was analyzed by analysis of variance using the Statistical Analysis System software package, SAS 9.4 (SAS Institute, Cary, NC, USA). Duncan’s multiple range test was used to determine the significance of differences among the means at the 0.05 level.

## 3. Results and Discussion

This study utilized various ratios of binary extraction solvents (water and ethanol) to extract bioactive compounds from different parts of tamarillo. The extract yield and bioactive compound content were determined. The extraction conditions yielding the highest bioactive compound content were selected for further investigation, including identifying individual bioactive compounds and their antioxidant activity and inhibitory effects on enzymes related to MetS prevention. Furthermore, to evaluate the impact of gastrointestinal digestion on the biochemical composition and bioactivity of the extracts, an in vitro gastrointestinal digestion simulation was conducted. This approach provides a more comprehensive understanding of the potential physiological effects of different tamarillo components after digestion in the human gastrointestinal tract. Ultimately, this study aims to determine the suitable processing methods to maximize the bioavailability of tamarillo as a functional supplement, ensuring its efficacy in preventing MetS.

### 3.1. The Yield of Extraction

The diverse structures of bioactive compounds, along with their varying polarity and solubility, can complicate the extraction process [44]. Therefore, the extraction method must be carefully selected to ensure food-grade safety and environmental sustainability. In the food industry, the choice of extraction solvents should ensure high efficiency and comply with safety and sustainability standards, making green extraction solvents a preferred option [45]. Moreover, a single solvent is often insufficient to efficiently extract all bioactive compounds from plant materials. As a result, binary solvent systems, such as water–organic solvent mixtures, are commonly employed to enhance the extraction of both polar and non-polar compounds, thereby improving overall yield and efficiency [46].

This study extracted freeze-dried tamarillo powder from different fruit parts using water and ethanol in varying ratios (0%, 25%, 50%, 75%, and 95% ethanol). The extraction yield (%) was calculated by dividing the weight of the freeze-dried extract by the weight of the original sample and multiplying the result by 100. As shown in Appendix A, the extraction yield of different tamarillo parts decreased as ethanol concentration increased. When extracted with water alone (0E), peel, pulp, mucilage, seeds, and whole fruit yielded 26.72%, 65.16%, 92.86%, 17.81%, and 57.64%, respectively. However, as the ethanol concentration increased to 95% (95E), the extraction yields decreased to 11.21%, 41.17%, 56.49%, 6.67%, and 33.68%, respectively.

### 3.2. Bioactive Compounds in Freeze-Dried Tamarillo Extracts

The total phenols, anthocyanins, and carotenoids content in tamarillo extracts from different fruit parts obtained using various water-to-ethanol ratios were analyzed, and the results are presented in Figure 1. Unlike the extraction yield, which decreased as ethanol concentration increased, the concentrations of these three groups of bioactive compounds exhibited an increasing trend with higher ethanol ratios.

#### 3.2.1. Total Phenols in Tamarillo Extracts

The total phenols content of freeze-dried tamarillo extracts from different fruit parts is presented in Figure 1A. Except for seeds, which exhibited the highest total phenol content at 75% ethanol (75E, 48.31 mg GAE/g freeze-dried extract), all other fruit parts showed the highest levels at 95% ethanol (95E). The total phenol content followed the order of peel (153.0 mg GAE/g) > whole fruit (34.64) > pulp (21.19) > mucilage (19.85).

#### 3.2.2. Total Anthocyanins in Tamarillo Extracts

As the ethanol concentration in the extraction solvent increased, the total anthocyanins content in the freeze-dried extracts of different tamarillo parts also increased, with the highest levels generally observed at 95E (Figure 1B). The total anthocyanin content (mg C3GE/g freeze-dried extract) was of the following order: mucilage (8.140) > peel (5.600) > whole fruit (3.674) > seeds (2.195) > pulp (1.110).

#### 3.2.3. Total Carotenoids in Tamarillo Extracts

As shown in Figure 1C, the total carotenoids content in freeze-dried tamarillo extracts increased with higher ethanol concentrations, with the highest levels generally observed at 95E, except for seeds, which exhibited the highest content at 75E (0.982 mg βCE/g freeze-dried extract). At 95E, the total carotenoid content (mg βCE/g freeze-dried extract) in different fruit parts was in the descending order of peel (4.324) > pulp (1.174) > whole fruit (0.771) > mucilage (0.466).

#### 3.2.4. Extraction Conditions

Based on the experimental results, 95% ethanol was the most suitable extract for freeze-dried tamarillo peel, pulp, mucilage, and whole fruit, yielding the highest levels of bioactive compounds. In contrast, 75% ethanol was more effective for extracting bioactive compounds from seeds. Therefore, tamarillo extracts obtained using 95% ethanol for peel (P95E), pulp (L95E), mucilage (M95E), and whole fruit (W95E), as well as the extract obtained using 75% ethanol for seeds (S75E), were selected for further analysis to evaluate their antioxidant capacity and enzyme-inhibitory effects related to MetS prevention.

### 3.3. Individual Bioactive Compounds

#### 3.3.1. Hydroxycinnamoyl Derivatives

According to a study on the phenolic composition of Ecuadorian tamarillo, the major hydroxycinnamoyl derivatives identified were 3-o-caffeoylquinic acid (chlorogenic acid) and rosmarinic acid [30]. Based on these findings, chlorogenic acid and rosmarinic acid were selected for analysis in this present study. The chlorogenic acid content (mg/g freeze-dried extract) in the freeze-dried extracts of different tamarillo parts followed the order of P95E (74.90) > W95E (9.234) > L95E (5.707) > S75E (1.868) ≈ M95E (1.696) (Table 1). Similarly, rosmarinic acid content (mg/g freeze-dried extract) was significantly higher in P95E (66.69) than in other parts, followed by W95E (6.697), with the lowest content observed in M95E (0.242). When combining chlorogenic acid and rosmarinic acid, the total hydroxycinnamoyl derivatives content (mg/g freeze-dried extract) was highest in P95E (141.6), followed by W95E (15.93) > L95E (8.934) > S75E (3.412) > M95E (1.938).

When recalculating the total hydroxycinnamoyl derivatives content in the freeze-dried samples based on the extraction yield, the following formula was applied:Total hydroxycinnamoyl derivatives content (mg/g freeze-dried sample) =content in freeze-dried extract × corresponding extraction yield %/100

Using this calculation, the total hydroxycinnamoyl derivatives content in P95E, L95E, M95E, S75E, and W95E was determined to be 15.87, 3.678, 1.094, 0.2146, and 5.365 mg/g of the freeze-dried sample, respectively. Notably, the whole fruit extract (536.5 mg/100 g freeze-dried sample) exhibited a higher total hydroxycinnamoyl derivatives content compared to the Giant Purple cultivar (132.6 ± 0.909 mg/100 g dry weight) and the New Zealand Purple cultivar (421.6 ± 3.082 mg/100 g dry weight) of tamarillo, as reported by Espín et al. (2016) [30].

#### 3.3.2. Anthocyanins

The individual anthocyanin content in freeze-dried extracts of different tamarillo parts is presented in Table 1. Among mucilage, seeds, and whole fruit, delphinidin-3-O-rutinoside was the most abundant anthocyanin, with 6.606, 4.179, and 2.186 mg/g freeze-dried extract, respectively. In contrast, cyanidin-3-O-rutinoside was predominant in the peel (8.687 mg/g freeze-dried extract), while pelargonidin-3-O-rutinoside was most abundant in the mucilage (6.111 mg/g freeze-dried extract). Regarding total anthocyanins content (mg/g freeze-dried extract), mucilage exhibited the highest levels (13.39), followed by peel (10.64) > seeds (7.630) > whole fruit (5.005) > pulp (0.625). Anthocyanins in pulp and seeds may be attributed to residual mucilage. When recalculated based on extraction yield (using the same method described in Section 3.3.1), the total anthocyanins content (mg/g freeze-dried sample) in P95E, L95E, M95E, S75E, and W95E was 1.193, 0.257, 7.560, 0.480, and 1.686, respectively. These values were higher than those reported by Espín et al. (2016) [30] for the purple tamarillo cultivars from Ecuador and New Zealand.

#### 3.3.3. Carotenoids

The carotenoids content in tamarillo extracts is shown in Table 1. The peel extract contained the highest levels of β-cryptoxanthin (0.928) and β-carotene (0.942). β-cryptoxanthin was also the predominant carotenoid in all other parts, with concentrations (mg/g freeze-dried extract) following the order of L95E (0.240) ≈ W95E (0.238) > M95E (0.085) > S75E (0.061). Additionally, only β-cryptoxanthin and β-carotene were detected in M95E, whereas S75E contained β-cryptoxanthin and zeaxanthin. Notably, lycopene and α-carotene were not detected in any extract. The total carotenoids content (mg/g freeze-dried extract) followed the order of P95E (1.943) > L95E (0.431) > W95E (0.318) > M95E (0.136) > S75E (0.069). These results align with the findings from Refs. [47,48], which reported that β-cryptoxanthin and β-carotene are the primary carotenoids in tamarillo.

### 3.4. Antioxidation Properties

The antioxidant capacity of tamarillo extracts was evaluated using two commonly applied in vitro assays: DPPH radical-scavenging activity and reducing power. The DPPH assay measures the ability of antioxidants to donate hydrogen atoms or electrons to neutralize DPPH free radicals [49] while the reducing power assay reflects the electron-donating capacity of the sample [50]. In both methods, the EC_50_ value, defined as the extract concentration required to achieve 50% of maximum activity, was used as an indicator of antioxidant strength, with lower EC_50_ values indicating stronger antioxidant potential. Given that oxidative stress is a key contributor to the development of metabolic syndrome [13], the antioxidant performance of tamarillo extracts may play an important role in their potential preventive effects against MetS.

#### 3.4.1. DPPH Radical-Scavenging Ability

The DPPH radical-scavenging capacity of freeze-dried tamarillo extracts from different fruit parts and whole fruit is presented in Appendix A. As the extract concentration increased, the DPPH scavenging activity also increased. At a low concentration (50 μg/mL), the scavenging capacity followed the order of P95E (55.24%) > S75E (17.74%) > W95E (14.30%) > L95E (10.09%) > M95E (8.66%). P95E exhibited the highest scavenging activity, exceeding 90% at 100 μg/mL. The EC_50_ values (μg extract/mL) for the DPPH scavenging activity are presented in Table 2. The lowest EC_50_ value (μg extract/mL), indicating the strongest antioxidant activity, was observed in P95E (45.26), followed by S75E (181.6) > W95E (221.0) > L95E (388.2) ≈ M95E (386.4). To account for differences in phenolic concentration across samples, EC_50_ values were recalculated based on total phenolic content (μg total phenols/mL), allowing for a more accurate comparison of antioxidant efficiency per unit of active compound. The recalculated EC_50_ values were P95E (9.420) > S75E (10.27) > W95E (12.24) > M95E (14.93) > L95E (15.57). Overall, tamarillo extract showed a strong DPPH radical-scavenging ability compared to standard antioxidant ascorbic acid (15.60 μg/mL), BHA (17.51 μg/mL), and α-tocopherol (16.35 μg/mL).

#### 3.4.2. Reducing Power

The reducing power of freeze-dried tamarillo extracts from different fruit parts and whole fruit is presented in Appendix A. As the extract concentration increased, the reducing power also increased. At a low concentration (200 μg/mL), the reducing power (absorbance at 700 nm) followed the order of P95E (0.873 AU) > W95E (0.179 AU) ≈ S75E (0.175 AU) > M95E (0.119 AU) ≈ L95E (0.117 AU). The EC_50_ values for the reducing power assay are presented in Table 2. The lowest EC_50_ value (μg extract/mL), indicating the highest reducing capacity, was observed in P95E (113.3 μg/mL), followed by S75E (597.2 μg/mL) > W95E (618.8 μg/mL) > M95E (938.4 μg/mL) > L95E (956.7 μg/mL). To further evaluate the reducing power relative to phenolic concentration, EC_50_ values were recalculated based on the total phenolic content (μg total phenols/mL), providing a normalized indicator of antioxidant efficiency per unit of phenolics. The recalculated EC_50_ values followed the order of P95E (23.58) > S75E (33.77) > W95E (34.27) > M95E (36.26) > L95E (38.37). All extracts’ reduced capacity (EC50 value) was significantly stronger than α-tocopherol (118.3 μg/mL). Additionally, P95E, S75E, and W95E exhibited stronger reducing power than BHA (34.98 μg/mL), though all extracts showed a lower reducing capacity than ascorbic acid (19.82 μg/mL).

The results of this study demonstrate that tamarillo extracts exhibit strong antioxidant capacity, as evidenced by their DPPH radical-scavenging activity and reducing power. Among the tested extracts, P95E consistently exhibited the highest antioxidant potential among the tested extracts, with the lowest EC_50_ values in both assays, followed by S75E and W95E. This antioxidant activity was positively correlated with total phenolic content, as P95E contained the highest level of phenolic compounds among all tamarillo fractions, followed by S75E and W95E. According to Del-Toro-Sánchez et al. (2021), phenolic compounds, particularly flavonoids, tend to be hydrophobic due to the presence of aromatic A and B rings [51]. This observation is consistent with the findings in Figure 1 of the present study, where increasing ethanol concentration and solvent hydrophobicity enhanced the extraction efficiency of total phenolics.

Moreover, in alcoholic extracts, flavonoids primarily exert their antioxidant effects through hydrogen atom donation. These compounds neutralize free radicals via redox reactions by donating hydrogen atoms (H-atoms), a mechanism that contributes significantly to their radical-scavenging activity [51,52]. This mechanism supports the strong antioxidant potential observed in tamarillo extracts, particularly those obtained using higher ethanol concentrations. Given the well-established link between OS and inflammation-related diseases [20,53], these findings suggest that tamarillo extracts may possess anti-inflammatory properties. Moreover, since OS is a key factor in developing metabolic syndrome, the potent antioxidant activity observed in tamarillo extracts highlights their potential as functional food ingredients for MetS prevention.

### 3.5. Enzyme-Inhibitory Activity

Hyperlipidemia, hyperglycemia, and hypertension are key indicators of MetS. Therefore, this study evaluated the ability of tamarillo extracts from different parts of the fruit to inhibit key enzymes associated with these conditions, including pancreatic lipase (involved in fat digestion and absorption), α-amylase and α-glucosidase (involved in carbohydrate hydrolysis), and ACE (related to blood pressure regulation). Furthermore, the inhibitory effects of tamarillo extracts were compared with commercially available medications to assess their potential in preventing MetS.

#### 3.5.1. Pancreatic Lipase Inhibitory Activity

Pancreatic lipase plays an important role in fat digestion and absorption, and its inhibition has been linked to potential anti-obesity effects [54]. Polyphenols, anthocyanins, and carotenoids have been reported to inhibit pancreatic lipase activity, thereby contributing to weight management [55,56,57]. This study evaluated the pancreatic lipase inhibitory capacity of freeze-dried tamarillo extracts derived from different fruit parts and the whole fruit. Except for the seed extract, all tamarillo extracts exhibited a dose-dependent inhibition of pancreatic lipase, with inhibition increasing as extract concentration increased (Appendix A). At a low concentration (1 mg/mL), the inhibitory effects of the tamarillo extracts followed the order of L95E (56.68%) > M95E (34.57%) > W95E (31.75%) > P95E (6.390%) > S75E (0%). At 3 mg/mL, L95E achieved 90% inhibition, whereas M95E and W95E required a higher concentration of 4 mg/mL to surpass 90% inhibition. In contrast, S75E exhibited no inhibition at 5 mg/mL; even at 20 mg/mL, its inhibition rate remained relatively low at 59.81% (Appendix A). The IC_50_ values (the concentration required to inhibit 50% of enzyme activity) for tamarillo extracts and orlistat are presented in Table 3, with inhibitory potency ranking as follows: Orlistat (9.066 ng/mL) > L95E (0.882 mg/mL) > M95E (1.637 mg/mL) > W95E (1.861 mg/mL) > P95E (2.964 mg/mL) > S75E (16.67 mg/mL). These findings indicate that orlistat exhibits significantly higher pancreatic lipase inhibitory activity than tamarillo extracts. However, the consumption of orlistat has been associated with adverse side effects such as fecal incontinence, bloating, and steatorrhea [58].

#### 3.5.2. α-Amylase and α-Glucosidase Inhibitory Activity

α-Amylase and α-glucosidase are key carbohydrate-digesting enzymes associated with postprandial hyperglycemia in individuals with type 2 diabetes. α-Amylase catalyzes the cleavage of α-1,4-glycosidic bonds, breaking down polysaccharides into smaller oligosaccharide fragments, while α-glucosidase further hydrolyzes these oligosaccharides into glucose. Inhibiting these enzymes can slow carbohydrate digestion and absorption, thereby helping to regulate postprandial blood glucose levels [59,60,61]. Additionally, phenolic acids, anthocyanins, and carotenoids have been reported to inhibit α-amylase and α-glucosidase activities, suggesting their potential role in glycemic control [62,63,64].

The α-amylase inhibitory activity of freeze-dried tamarillo extracts is presented in Appendix A. The inhibition rate increased with extract concentration, demonstrating a dose-dependent effect. At 1 mg/mL, the α-amylase inhibition of tamarillo extracts followed the order of W95E (6.419%) > L95E (5.387%) ≈ M95E (4.954%) > S75E (2.966%) > P95E (1.685%). At 3 mg/mL, L95E exhibited over 90% inhibition of α-amylase activity. For W95E and M95E, the concentrations required to achieve 90% inhibition were 4 mg/mL and 5 mg/mL, respectively. P95E showed a rapid increase in α-amylase inhibition between 5 mg/mL and 6 mg/mL, rising from 27.32% to 89.02% (Appendix A). Meanwhile, S75E required 15 mg/mL and 20 mg/mL concentrations to reach inhibition rates of 47.12% and 86.91%, respectively (Appendix A). The IC_50_ values for tamarillo extracts and acarbose, a commercial α-amylase inhibitor, are shown in Table 3. The inhibitory potency followed the order of acarbose (5.525 μg/mL) > L95E (2.369 mg/mL) > W95E (3.132 mg/mL) > M95E (3.351 mg/mL) > P95E (5.368 mg/mL) > S75E (14.80 mg/mL). These results indicate that acarbose exhibits significantly stronger α-amylase inhibitory activity than tamarillo extracts.

The α-glucosidase inhibitory activity of freeze-dried tamarillo extracts from different fruit parts and the whole fruit is shown in Appendix A. The inhibitory effect increased with extract concentration, demonstrating a dose-dependent response. At a low concentration (2 mg/mL), the inhibition rates of tamarillo extracts followed the order of P95E (61.62%) > S75E (37.48%) > W95E (17.00%) > L95E (2.792%) > M95E (0%). At 10 mg/mL, the inhibition rates for different tamarillo parts were as follows: peel (96.80%) > seeds (75.31%) > whole fruit (72.50%) > pulp (70.35%) > mucilage (57.33%). The IC_50_ values (Table 3) representing the concentration required to inhibit 50% of α-glucosidase activity followed the order of acarbose (91.21 μg/mL) > P95E (1.623 mg/mL) > S75E (3.581 mg/mL) > W95E (5.534 mg/mL) > L95E (6.299 mg/mL) > M95E (8.856 mg/mL).

Acarbose is a commonly prescribed drug for type 2 diabetes that inhibits both α-amylase and α-glucosidase, slowing glucose absorption and reducing postprandial blood sugar spikes. Indeed, the IC_50_ values of acarbose for inhibiting α-amylase and α-glucosidase were significantly lower than those of tamarillo extracts, indicating its superior inhibitory potency. However, acarbose consumption has been associated with adverse side effects, such as abdominal discomfort, diarrhea, bloating, and hepatotoxicity [65,66].

#### 3.5.3. ACE Inhibitory Activity

The renin–angiotensin system regulates blood pressure, cardiac function, and vascular homeostasis. Renin catalyzes the conversion of angiotensinogen into angiotensin I, which is subsequently cleaved by ACE to generate angiotensin II, a potent vasoconstrictor. Inhibiting ACE activity is a well-established strategy for managing hypertension [67]. Polyphenols, anthocyanins, and carotenoids have been reported to contribute to ACE inhibition, demonstrating potential antihypertensive effects [68,69,70].

The ACE inhibitory activity of freeze-dried tamarillo extracts is presented in Appendix A. The inhibition rate increased with extract concentration, exhibiting a dose-dependent effect. At a low concentration (1 mg/mL), the ACE inhibition rates followed the order of P95E (42.73%) ≈ S75E (42.65%) > W95E (26.46%) ≈ M95E (24.58%) > L95E (15.39%). At 6 mg/mL, the inhibition rates increased to P95E (87.12%) > S75E (83.26%) ≈ W95E (81.05%) ≈ M95E (78.47%) ≈ L95E (75.40%). The IC_50_ values (Table 3), representing the concentration required to inhibit 50% of ACE activity, followed the order of captopril (0.481 ng/mL) > P95E (1.435 mg/mL) ≈ S75E (1.417 mg/mL) > W95E (3.479 mg/mL) ≈ M95E (3.587 mg/mL) > L95E (4.038 mg/mL).

Captopril, a widely used antihypertensive drug, effectively inhibits ACE activity. As evidenced by the results, the IC_50_ value of captopril was significantly lower than those of tamarillo extracts, indicating its superior inhibitory potency. However, the use of captopril has been associated with adverse side effects, including chronic cough, taste disturbances, and skin rashes [71].

These findings suggest that freeze-dried tamarillo extracts have the potential for anti-obesity effects and the prevention of type 2 diabetes and hypertension. Although their inhibitory effects were not as strong as those of pharmaceutical drugs, concerns remain regarding the side effects associated with drug use. Consequently, there is growing interest in natural, food-based alternatives for inhibiting enzymes related to MetS, positioning tamarillo extracts as promising functional ingredients for its prevention.

In support of this potential, we observed that the α-glucosidase and ACE inhibitory activities were positively correlated with the total phenolic content. Notably, samples P95E and S75E, which contained higher levels of total phenols, exhibited stronger α-glucosidase and ACE inhibitory activities. In contrast, no positive correlation was found between pancreatic lipase or α-amylase inhibitory activities and the levels of total phenols or other phytochemicals. Interestingly, samples L95E and M95E showed the highest inhibitory activities against pancreatic lipase and α-amylase, despite containing relatively low levels of phytochemicals in this study.

These observations suggest that the inhibitory mechanisms of phytochemicals may vary depending on the enzyme involved. Phytochemicals can interact with digestive enzymes through either non-competitive or competitive inhibition. For example, flavonoids have been shown to act via non-competitive inhibition in α-glucosidase while certain polyphenols may competitively bind to the active site of α-amylase [72]. These interactions likely involve complex binding mechanisms, including hydrogen bonding and other non-covalent forces, which can influence enzyme kinetics [72,73]. However, current studies on phytochemical–enzyme complexes remain insufficient to establish a comprehensive structure–activity relationship [73]. The conformational changes that occur in enzymes upon binding with different phytochemicals are expected to depend on the structural characteristics of both components. Therefore, further investigation is warranted to fully elucidate the specific binding mechanisms between digestive enzymes and phytochemicals.

### 3.6. Bioactive Compounds After Simulated In Vitro Gastrointestinal Digestion

Based on the findings of this study, tamarillo extracts are rich in bioactive compounds and exhibit strong inhibitory activity against enzymes associated with MetS. However, upon ingestion, these bioactive compounds undergo gastrointestinal digestion, during which they may be modified or degraded by digestive enzymes [74]. Therefore, understanding the effects of simulated digestion on these compounds’ bioactivity, bioaccessibility, and bioavailability is essential for further research, particularly in developing strategies to enhance their stability and functionality [75]. This knowledge can also support the development of targeted delivery techniques, facilitating the formulation of functional ingredients and determining effective dosages [76]. In this study, freeze-dried tamarillo extracts were subjected to simulated in vitro gastrointestinal digestion to evaluate changes in the content and bioactivity of bioactive compounds before and after digestion. Following the digestion simulation, the supernatant was collected as the soluble fraction, while the precipitate represented the insoluble portion. The dialysate inside the dialysis bag was also collected for analysis, as this fraction represents the compounds that have entered the bloodstream. This fraction is crucial, as it defines the bioavailability of the compounds, indicating their potential to exert physiological functions [77,78].

#### 3.6.1. Total Polyphenols in Digested Extracts

The total phenol content (mg GAE/200 mg freeze-dried extract) of tamarillo extracts before and after simulated in vitro gastrointestinal digestion is presented in Table 4. Among the intestinal supernatants (soluble fraction), P95E exhibited the highest total phenol content (34.32), followed by S75E (11.87) > W95E (10.28) > L95E (7.188) ≈ M95E (7.113). Notably, the total phenol content in the soluble fraction was significantly increased compared to the non-digested extracts. In contrast, the intestinal precipitate (insoluble fraction) showed a marked decrease in total phenol content relative to the non-digested extracts, with S75E having the highest content at 1.541 mg GAE, representing only 15.95% of its pre-digestion level. Similar observations have been reported in the literature. For instance, He et al. (2017) found that the total phenol content increased after digestion in 22 types of fruit juices [79], while Hwang et al. (2023) noted that cherry tomatoes maintained their phenol content during gastric digestion but experienced a significant reduction following intestinal digestion [80]. These findings underscore the sensitivity of phenolic compounds to the gastrointestinal environment. Phenolic compounds are highly responsive to pH changes; they tend to remain stable under mildly acidic conditions, such as those in the stomach but are prone to degradation in the more alkaline conditions of the small intestine [81]. This degradation may be further influenced by interactions between hydrolyzed polyphenols and proteins and the conversion of certain polyphenols into their respective breakdown products during intestinal digestion [82,83]. The study results indicate that the total phenol content in the dialysate significantly decreased after simulated gastrointestinal digestion. The total phenol content (mg GAE/200 mg freeze-dried extract) in the dialysate for P95E, L95E, M95E, S75E, and W95E was 3.168, 0.793, 0.746, 1.146, and 1.025, respectively. During the dialysis phase, only 10.36–18.79% of the total phenolic compounds could pass through the dialysis membrane, suggesting that only a small fraction of polyphenols can reach the bloodstream. The permeability of polyphenols through the dialysis membrane depends on several factors, including molecular size, degree of polymerization, and the presence of sugar moieties. Generally, monomeric or glycosylated derivatives are more likely to pass through the membrane than their polymerized counterparts [84]. These findings highlight the potential challenges in the bioavailability of polyphenols and underscore the importance of further research into strategies for enhancing their absorption and physiological effectiveness.

#### 3.6.2. Total Anthocyanins in Digested Extracts

The health benefits of anthocyanins depend highly on their bioavailability after digestion, which is influenced by their source, dosage, and the experimental model. Numerous studies on anthocyanin-rich food consumption have demonstrated that only a small fraction of ingested anthocyanins are absorbed intact into the bloodstream or excreted in the urine [85]. After oral ingestion, anthocyanins remain stable in the stomach’s acidic environment (pH 1.5–5.0), where some may be absorbed. However, upon entering the intestine, exposure to higher pH levels (pH 5.6–7.9), enzymatic activity, and gut microbiota metabolism contribute to their extensive degradation, further limiting their absorption into the bloodstream [85,86]. As a result, a substantial portion of anthocyanins is lost during digestion, significantly reducing their bioavailability. Our findings (Table 4) further confirm the significant decline in the total anthocyanins content after digestion. In the intestinal supernatant, detectable levels of total anthocyanins were observed only in P95E and M95E, with concentrations of 0.589 mg C3GE/200 mg freeze-dried extract and 1.406 mg C3GE/200 mg freeze-dried extract, corresponding to 52.59% and 86.36% of their pre-digestion content, respectively. However, no anthocyanins were retained in the intestinal precipitate, indicating their complete degradation or solubilization during digestion. Additionally, anthocyanin permeability through the dialysis membrane, which simulates potential absorption into the bloodstream, was detected in P95E, L95E, M95E, S75E, and W95E, with concentrations of 0.176, 0.021, 0.267, 0.063, and 0.068 mg C3GE/200 mg freeze-dried extract, respectively. This suggests that only 9.25% to 16.40% of the anthocyanins were potentially bioavailable for systemic circulation. Similarly, Yamazaki et al. reported that only 10–20% of anthocyanins are actively transported into the bloodstream through gastric epithelial cells, primarily in the form of their metabolites [87]. These findings highlight the substantial loss of anthocyanins during digestion and their limited bioavailability, emphasizing the need for strategies to enhance their stability and absorption.

#### 3.6.3. Total Carotenoids in Digested Extracts

The total carotenoid content of tamarillo extracts before and after in vitro gastrointestinal digestion simulation is presented in Table 4. In the intestinal supernatant, P95E exhibited the highest carotenoids content (0.435 mg βCE/200 mg freeze-dried extract), retaining 50.29% of its pre-digestion level, whereas S75E had the lowest content (0.038 mg βCE/200 mg freeze-dried extract), retaining only 20.43%. In the intestinal precipitate, the carotenoid content across different tamarillo extracts ranged from 0.003 to 0.067 mg βCE/200 mg freeze-dried extract, accounting for only 1.95–7.75% of the initial amount. Furthermore, only 9.36–17.53% of the total carotenoids from P95E, L95E, M95E, S75E, and W95E could pass through the dialysis membrane, which simulates potential absorption into the bloodstream. The concentrations detected in the dialysate were 0.081, 0.024, 0.016, 0.020, and 0.027 mg βCE/200 mg freeze-dried extract. These findings indicate that only a small fraction of carotenoids is bioavailable after digestion. Carotenoids are extremely sensitive to various environmental factors, including light exposure, oxidation, and pH fluctuations. Acidic conditions, in particular, significantly accelerate their decomposition, whereas degradation tends to be lower at pH values ranging from 6.0 to 7.0 [80]. Therefore, the acidic environment of gastric juice may influence the stability and subsequent uptake of β-carotene [88]. These results align with our findings, demonstrating a significant decline in total carotenoid content following digestion simulation and confirming that only a limited portion of carotenoids are available for systemic circulation.

### 3.7. Enzyme-Inhibitory Activity of Dialysate After Simulated In Vitro Gastrointestinal Digestion

Phytochemicals are susceptible to chemical degradation when exposed to environmental factors such as pH fluctuations, heat, light, oxygen, and prooxidants [89]. Similarly, polyphenols from different plant sources or types exhibit varying compositional and bioactivity changes after simulated gastrointestinal digestion. These variations are influenced by multiple factors, including the inherent chemical structures of phytochemicals and the food matrix, which collectively impact their stability and bioavailability [90,91]. Hwang and Kim (2023) reported a significant reduction in the total polyphenols, flavonoids, and carotenoids, as well as the antioxidant capacity, in cherry tomatoes after in vitro gastrointestinal digestion [80]. However, other studies have shown that despite digestion, certain plant infusions still retained their antioxidant potential and the ability to inhibit α-amylase, α-glucosidase, and ACE, suggesting their potential antidiabetic and antihypertensive effects [90].

The IC_50_ values for inhibiting pancreatic lipase, α-amylase, α-glucosidase, and ACE in the dialysis phase of in vitro digested tamarillo extracts are presented in Table 5. The results indicate a significant increase in IC_50_ values across all enzyme assays, suggesting that digestion reduces the biochemical activity of tamarillo extracts. A comparison of different fruit parts revealed increases to varying degrees in IC_50_ values after digestion. P95E exhibited the highest increase (approximately 8.2–9.0 times), followed by S75E and W95E (6.5–6.9 times and 5.8–6.4 times, respectively). L95E and M95E showed the lowest increase (approximately 4.9–5.2 and 4.9–5.3 times, respectively). These findings suggest that the resistance of tamarillo extracts to digestive degradation varies among different fruit parts. The pulp and mucilage retained the highest biochemical activity, whereas the peel exhibited the lowest retention. Nevertheless, all extracts retained some degree of inhibition against pancreatic lipase, α-amylase, α-glucosidase, and ACE, which are key enzymes involved in obesity, hyperglycemia, and hypertension. These results highlight the potential of tamarillo extracts as functional ingredients, as they retain partial effectiveness in preventing metabolic syndrome even after digestion.

This study initially evaluated and selected the suitable extraction conditions for different tamarillo fruit parts using a water–ethanol solvent mixture, revealing that the extracts were rich in polyphenols, anthocyanins, and carotenoids. These bioactive compounds exhibited strong antioxidant activity and significant inhibitory effects on pancreatic lipase, α-amylase, α-glucosidase, and ACE, suggesting that tamarillo may offer potential benefits in preventing obesity, hyperglycemia, and hypertension, thereby contributing to the prevention of MetS.

Our findings align with previous studies, indicating that the total polyphenols, anthocyanins, and carotenoids and the biochemical activities in plants are affected by digestion to varying degrees. This study specifically analyzed the soluble, insoluble, and dialysis phases post-digestion. While most tamarillo extracts exhibited reduced total polyphenols, anthocyanins, and carotenoids after digestion, the soluble phase contained a higher total polyphenols content than the undigested extracts, suggesting improved polyphenols solubility during digestion.

Since the dialysis phase represents bioavailable compounds capable of entering the bloodstream, we further evaluated its ability to inhibit pancreatic lipase, α-amylase, α-glucosidase, and ACE. The results showed that the IC_50_ values for these enzymes were significantly higher than those of the undigested extracts, indicating that digestion and absorption reduce overall biochemical activity. Nevertheless, the extracts retained some inhibitory effects on these key enzymes associated with MetS, suggesting that they still hold potential for MetS prevention.

Nevertheless, although a significant reduction in total phenolic content and inhibitory activities against MetS-related enzymes was observed in the dialysis phase of the in vitro digestion, the relative ranking of enzyme-inhibitory activities among the different plant parts remained consistent with the pre-digestion results. In particular, the inhibitory trends against α-glucosidase and ACE were maintained, with samples P95E and S75E, which retained the highest phenolic content post-digestion, continuing to exhibit the strongest inhibitory activities.

This study underscores the importance of considering the effects of gastrointestinal digestion and systemic absorption on the bioactivity of plant-derived compounds. Food processing technologies such as microencapsulation should be explored to enhance their bioavailability and stability. These strategies could improve the functional efficacy of tamarillo extracts as potential dietary supplements or functional foods for metabolic syndrome prevention.

## 4. Conclusions

This study investigated tamarillo extracts’ phytochemical composition, antioxidant capacity, and enzyme-inhibitory effects, assessing the impact of varying ethanol concentrations in the extraction solvents and simulated gastrointestinal digestion. The results demonstrated that ethanol concentration significantly influenced the extraction efficiency of bioactive compounds, with 95% ethanol yielding the highest levels of polyphenols, anthocyanins, and carotenoids in most fruit parts. Tamarillo extracts exhibited strong antioxidant activity and notable inhibitory effects against key metabolic syndrome-related enzymes, including pancreatic lipase, α-amylase, α-glucosidase, and ACE. However, simulated gastrointestinal digestion reduced phytochemical content and enzyme-inhibitory activity, although some bioactivity remained in the dialysate fraction, indicating potential bioavailability. This result could serve as a reference for subsequent animal or human trials on tamarillo extracts. These findings highlight the functional potential of tamarillo as a natural source of bioactive compounds for MetS prevention. Future research should focus on enhancing the stability and bioavailability of tamarillo phytochemicals, such as through encapsulation technologies or food matrix integration, to maximize their health benefits in functional food applications.

## Figures and Tables

**Figure 1 foods-14-01282-f001:**
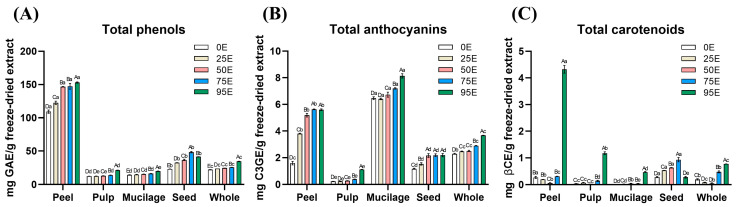
Bioactive compounds content of freeze-dried extracts. (**A**) Total phenols; (**B**) total anthocyanins; (**C**) total carotenoids. Each value is expressed as mean ± standard deviation (n = 3). Means with different capital letters from the same fruit parts extracted with different ratios of water–ethanol solvents differ significantly (*p* < 0.05). Means with different lowercase letters from the extracts of different fruit parts using the same solvent ratio differ significantly (*p* < 0.05).

**Table 1 foods-14-01282-t001:** Hydroxycinnamoyl derivatives, anthocyanins, and carotenoids content of freeze-dried extracts.

	P95E ^1^	L95E	M95E	S75E	W95E
Hydroxycinnamoyl derivatives (mg/g freeze-dried extract)					
Chlorogenic acid	74.90 ± 0.850 Aa ^2^	5.707 ± 0.082 Ac	1.696 ± 0.075 Ad	1.868 ± 0.064 Ad	9.234 ± 0.103 Ab
Rosmarinic acid	66.69 ± 0.090 Ba	3.227 ± 0.072 Bc	0.242 ± 0.001 Be	1.544 ± 0.043 Bd	6.697 ± 0.070 Bb
Total	141.6 ± 0.800 a	8.934 ± 0.154 c	1.938 ± 0.076 e	3.412 ± 0.106 d	15.93 ± 0.170 b
Anthocyanins (mg/g freeze-dried extract)					
Delphinidin-3-O-rutinoside	0.659 ± 0.006 Cd	0.253 ± 0.008 Be	6.606 ± 0.101Aa	4.179 ± 0.062 Ab	2.186 ± 0.004 Ac
Pelargonidin-3-O-rutinoside	1.291 ± 0.007 Bd	0.311 ± 0.001 Ae	6.111 ± 0.084 Ba	3.116 ± 0.151 Bb	2.055 ± 0.022 Bc
Cyanidin-3-O-rutinoside	8.687 ± 0.116 Aa	0.061 ± 0.007 Cd	0.675 ± 0.069Cb	0.336 ± 0.013 Cc	0.763 ± 0.001 Cb
Total	10.64 ± 0.120 b	0.625 ± 0.014 e	13.39 ± 0.086 a	7.630 ± 0.077 c	5.005 ± 0.027 d
Carotenoids (mg/g freeze-dried extract)					
Zeaxanthin	0.073 ± 0.003 Ba ^2^	0.006 ±< 0.001 Cc	nd	0.009 ±< 0.001Bbc	0.014 ± 0.004 Cb
β-Cryptoxanthin	0.928 ± 0.020 Aa	0.240 ±< 0.001 Ab	0.085 ± 0.006 Ac	0.061 ±< 0.001 Ad	0.238 ± 0.004 Ab
Lycopene	nd^3^	nd	nd	nd	nd
α-Carotene	nd	nd	nd	nd	nd
β-Carotene	0.942 ± 0.020 Aa	0.184 ± 0.008 Bb	0.051 ± 0.006 Bc	nd	0.066 ± 0.005 Bc
Total	1.943 ± 0.044 a	0.431 ± 0.009 b	0.136 ± 0.012 d	0.069 ± 0.001 e	0.318 ± 0.005 c

^1^ P95E, L95E, M95E, and W95E: the freeze-dried extracts were prepared from freeze-dried tamarillo peel, pulp, mucilage, and whole fruit with 95% ethanol at 75 °C in a shaking water bath (150 rpm), respectively. S75E, the freeze-dried extracts were prepared from freeze-dried tamarillo seed with 75% ethanol at 75 °C in a shaking water bath (150 rpm). ^2^ Each value is expressed as mean ± standard deviation (n = 3). Means with different capital letters within a column differ significantly (*p* < 0.05). Means with different lowercase letters within a row differ significantly (*p* < 0.05).

**Table 2 foods-14-01282-t002:** EC_50_ values of freeze-dried extracts for scavenging ability on 2,2-diphenyl-1-picrylhydrazyl radicals and reducing power.

Extraction Method	EC_50_ Value of Scavenging Ability ^2^		EC_50_ Value of Reducing Power ^3^	
	Extract (μg extract/mL)	Extract (μg Total phenols/mL)	Extract (μg extract/mL)	Extract (μg Total phenols/mL)
P95E ^1^	45.26 ± 0.11 D ^4^	9.420 ± 0.18 E	113.3 ± 1.0 E	23.58 ± 0.72 D
L95E	388.2 ± 4.5 A	15.57 ± 0.42 A	956.7 ± 8.2 A	38.37 ± 0.91 A
M95E	386.4 ± 4.2 A	14.93 ± 0.05 B	938.4 ± 5.1 B	36.26 ± 0.47 B
S75E	181.6 ± 2.3 C	10.27 ± 0.06 D	597.2 ± 1.4 D	33.77 ± 0.69 C
W95E	221.0 ± 2.7 B	12.24 ± 0.04 C	618.8 ± 4.3 C	34.27 ± 0.29 C

^1^ P95E, L95E, M95E, and W95E: the freeze-dried extracts were prepared from freeze-dried tamarillo peel, pulp, mucilage, and whole fruit with 95% ethanol at 75 °C in a shaking water bath (150 rpm), respectively. S75E, the freeze-dried extracts were prepared from freeze-dried tamarillo seed with 75% ethanol at 75 °C in a shaking water bath (150 rpm). ^2^ EC_50_ value of scavenging ability: the effective concentration at which 2,2-diphenyl-1-picrylhydrazyl (DPPH) radicals were scavenged by 50%. EC_50_ value was obtained by interpolation from linear regression analysis. EC_50_ values of ascorbic acid, BHA, and α-tocopherol were 15.60 ± 0.14, 17.51 ± 0.12, and 16.35 ± 0.10 μg/mL, respectively. ^3^ EC_50_ value of reducing power: the effective concentration at which the absorbance is 0.5. EC_50_ values of ascorbic acid, BHA, and α-tocopherol were 19.87 ± 0.03, 34.98 ± 0.55, and 118.31 ± 0.43 μg/mL, respectively. ^4^ Each value is expressed as mean ± standard deviation (n = 3). Means with different capital letters within a column differ significantly (*p* < 0.05).

**Table 3 foods-14-01282-t003:** IC_50_ values of inhibitory effects of freeze-dried extracts against pancreatic lipase, α-amylase, α-glucosidase, and angiotensin-converting enzymes.

Extraction Method ^1^	IC_50_ Values (mg/mL) ^2^			
	Pancreatic Lipase	α-Amylase	α-Glucosidase	ACE ^3^
P95E	2.964 ± 0.032 B ^4^	5.368 ± 0.002 B	1.623 ± 0.019 E	1.435 ± 0.103 D
L95E	0.882 ± 0.007 E	2.369 ± 0.006 E	6.299 ± 0.109 B	4.038 ± 0.091 A
M95E	1.637 ± 0.002 D	3.351 ± 0.029 C	8.856 ± 0.117 A	3.587 ± 0.173 B
S75E	16.67 ± 0.290 A	14.80 ± 0.020 A	3.581 ± 0.106 D	1.471 ± 0.049 D
W95E	1.861 ± 0.030 C	3.132 ± 0.004 D	5.534 ± 0.102 C	3.349 ± 0.196 C
Orlistat	(5.525 ± 0.060) × 10^−3^ F	-	-	-
Acarbose	-	(9.121 ± 0.069) × 10^−2^ F	(9.121 ± 0.069) × 10^−2^ F	-
Captopril	-	-	-	(0.481 ± 0.003) × 10^−6^ E

^1^ P95E, L95E, M95E, and W95E: the freeze-dried extracts were prepared from freeze-dried tamarillo peel, pulp, mucilage, and whole fruit with 95% ethanol at 75 °C in a shaking water bath (150 rpm), respectively. S75E, the freeze-dried extracts were prepared from freeze-dried tamarillo seed with 75% ethanol at 75 °C in a shaking water bath (150 rpm). ^2^ IC_50_ value: the enzyme activity was inhibited by 50%. The IC_50_ value was obtained by interpolation from linear regression analysis. ^3^ ACE: angiotensin-converting enzyme. ^4^ Each value is expressed as mean ± standard deviation (n = 3). Means with different capital letters within a column differ significantly (*p* < 0.05).

**Table 4 foods-14-01282-t004:** Amount of total phenols, anthocyanins, and carotenoids in the tested 200 mg of tamarillo extracts before (non-digested original) and after the simulated in vitro gastrointestinal digestion.

	P95E ^4^	L95E	M95E	S75E	W95E
Total phenols (mg GAE ^1^/200 mg freeze-dried extract)					
Non-digested original	30.59 ± 0.175 a ^5^	4.237 ± 0.030 d	3.970 ± 0.062 e	9.662 ± 0.210 b	6.928 ± 0.009 c
Sampled from intestinal phase (soluble)	34.32 ± 0.219 a	7.188 ± 0.153 d	7.113 ± 0.020 d	11.87 ± 0.270 b	10.28 ± 0.194 c
Sampled from intestinal phase (insoluble)	1.174 ± 0.033 b	1.115 ± 0.006 bc	1.115 ± 0.003 bc	1.541 ± 0.059 a	1.103 ± 0.009 c
Sampled from dialysis phase	3.168 ± 0.294 a	0.793 ± 0.010 c	0.746 ± 0.037 c	1.146 ± 0.033 b	1.025 ± 0.027 bc
Total anthocyanins (mg C3GE ^2^/200 mg freeze-dried extract)					
Non-digested original	1.120 ± 0.013 b	0.222 ± 0.002 e	1.628 ± 0.032 a	0.437 ± 0.021 d	0.735 ± 0.002 c
Sampled from intestinal phase (soluble)	0.589 ± 0.014 b	nd	1.406 ± 0.015 a	nd	nd
Sampled from intestinal phase (insoluble)	nd ^6^	nd	nd	nd	nd
Sampled from dialysis phase	0.176 ± 0.015 b	0.021 ± 0.008 d	0.267 ± 0.015 a	0.063 ± 0.013 c	0.068 ± 0.016 c
Total carotenoids (mg βCE ^3^/200 mg freeze-dried extract)					
Non-digested original	0.865 ± 0.029 a	0.235 ± 0.010 b	0.093 ± 0.004 e	0.186 ± 0.016 c	0.154 ±< 0.001 d
Sampled from intestinal phase (soluble)	0.435 ± 0.022 a	0.157 ± 0.021 b	0.049 ± 0.022 d	0.038 ± 0.012 d	0.082 ± 0.020 c
Sampled from intestinal phase (insoluble)	0.067 ± 0.029 a	0.016 ± 0.005 b	0.004 ± 0.001 b	0.004 ± 0.003 b	0.003 ± 0.002 b
Sampled from dialysis phase	0.081 ± 0.004 a	0.024 ± 0.010 b	0.016 ± 0.004 b	0.020 ± 0.015 b	0.027 ± 0.002 b

^1^ GAE꞉ Gallic acid equivalent. It estimated concentrations of 152.96, 21.19, 19.85, 48.31, and 34.64 mg of total phenol g^−1^ dry weight extract. ^2^ C3GE: cyanidin-3-O-glucoside equivalent. It estimated concentrations of 5.60, 1.11, 8.14, 2.18, and 3.67 mg of total anthocyanin g^−1^ dry weight extract. ^3^ βCE: β-carotene equivalent. It estimated concentrations of 4.32, 1.17, 0.47, 0.93, and 0.77 mg of total carotenoid g^−1^ dry weight extract. ^4^ P95E, L95E, M95E, and W95E: the samples were prepared from freeze-dried tamarillo peel, pulp, mucilage, and whole fruit extracts before and after in vitro gastrointestinal digestion simulation. ^5^ Each value is expressed as mean ± standard deviation (n = 3). Means with different lowercase letters within a row differ significantly (*p* < 0.05). ^6^ nd: not detectable.

**Table 5 foods-14-01282-t005:** Amounts of IC_50_ values of inhibition against pancreatic lipase, α-amylase, α-glucosidase, and angiotensin-converting enzymes in the tested 200 mg extracts before (non-digested original) and after the simulated in vitro gastrointestinal digestion (GD).

Sampled from Dialysis Phase ^1^		IC_50_ Values (mg/mL) ^2^			
		Pancreatic Lipase	α-Amylase	α-Glucosidase	ACE ^3^
P95E	Before GD	2.964 ± 0.032 B ^4^	5.368 ± 0.002 B	1.623 ± 0.019 E	1.435 ± 0.103 C
	After GD	24.72 ± 1.650 b	46.00 ± 0.170 b	13.23 ± 0.050 d	12.87 ± 0.890 b
L95E	Before GD	0.882 ± 0.0070 D	2.369 ± 0.006 E	6.299 ± 0.109 B	4.038 ± 0.091 A
	After GD	4.369 ± 0.077 d	12.42 ± 0.990 e	30.90 ± 0.990 b	20.45 ± 0.980 a
M95E	Before GD	1.637 ± 0.002 C	3.351 ± 0.029 C	8.856 ± 0.117 A	3.587 ± 0.173 B
	After GD	8.028 ± 0.187 cd	16.52 ± 0.270 d	44.88 ± 0.660 a	18.92 ± 1.750 a
S75E	Before GD	16.66 ± 0.290 A	14.80 ± 0.020 A	3.581 ± 0.106 D	1.471 ± 0.049 C
	After GD	109.1 ± 5.490 a	95.77 ± 3.210 a	24.76 ± 4.060 c	9.567 ± 0.612 c
W95E	Before GD	1.861 ± 0.030 C	3.132 ± 0.004 D	5.534 ± 0.102 C	3.349 ± 0.196 B
	After GD	11.44 ± 0.540 c	20.04 ± 0.730 c	32.00 ± 0.340 b	20.58 ± 2.140 a

^1^ The samples were in the dialysis phase after simulated digestion. ^2^ IC_50_ value: the enzyme activity was inhibited by 50%. The IC_50_ value obtained by interpolation from linear regression analysis. ^3^ ACE: angiotensin-converting enzyme. ^4^ Each value is expressed as mean ± standard deviation (n = 3). Means with different letters (a, b, c, d, e or A, B, C, D, E) within a column differ significantly (*p* < 0.05).

## Data Availability

The original contributions presented in this study are included in the article/Appendix A; further inquiries can be directed to the corresponding author.

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
