# Peer review of "Metabolic Syndrome Prevention Potential of Tamarillo: Phytochemical Composition, Antioxidant Activity, and Enzyme Inhibition Before and After Digestion"

_foods, 2025, doi:10.3390/foods14071282_

Round 1
Reviewer 1 Report
Comments and Suggestions for Authors
Dear Editor of Food MDPI, thank you for considering me as part of the review group for your prestigious journal. I have carefully reviewed the manuscript entitled "Metabolic Syndrome Prevention Potential of Tamarillo: Phytochemical Composition, Antioxidant Activity, and Enzyme Inhibition Before and After Digestion," and I consider the authors should make the following adjustments or respond to each comment.
Comment to authors
- Lines 12-15 should be treated as the objective of the research. You can start like this: the present research project was to evaluate different parts of the tomatillo....
- Lines 27-29 should be concluded regarding antioxidant activity, enzymatic activity and in vitro digestion in relation to metabolic syndrome.
- Lines 34-39, it should be defined what metabolic syndrome is and related to free radicals.
- The introduction should relate oxidative stress (OS) to Metabolic Syndrome. Furthermore, define an antioxidant: "a redox-active compound that limits oxidative stress by reacting non-enzymatically with a reactive oxidant.". Cite: (2018). Zein-polysaccharide nanoparticles as matrices for antioxidant compounds: A strategy for prevention of chronic degenerative diseases. Food Research International, 111, 451-471.
- In section 2.1 of materials and methods, an image of the tamarillo plant must be included, identifying the different parts that were evaluated in this research.
- Lines 453-455, the following is mentioned: "The results of this study demonstrate that tamarillo extracts exhibit strong antioxidant capacity, as evidenced by their DPPH radical-scavenging activity and reducing power", it should be defined how the individual compounds provide antioxidant activity, from the structural point of view, hydroxyls, double bonds, aromatic rings, etc. Cite: (2021). Recovery of phytochemical from three safflower (Carthamus tinctorius L.) by‐products: antioxidant properties, protective effect of human erythrocytes and profile by UPLC‐DAD‐Journal of Food Processing and Preservation, 45(9), e15765.
- In the results and discussion of in vitro digestion, it should be mentioned whether individual compounds undergo structural changes, dissociation, etc. and relate this to antioxidant activity, which decreases.
Comments on the Quality of English Language
The English presented is adequate, but I leave it to the academic editor's consideration whether language revision is necessary.
Reviewer 2 Report
Comments and Suggestions for Authors
This is an extensive and interesting manuscript focused on antioxidative and enzyme inhibitory properties of Tamarillo fruits. A lot of experimental work have been done, and the paper is well organized. In my opinion major corrections are needed, especially those related to experimental methodology. Using terms like "optimal" should be avoided unless a proper optimization study (e.g., response surface methodology) is conducted. No experimental design was reported to evaluate the influence of variables on extraction of phytochemicals. The authors only varied the content of ethanol in extraction solution, while other variables (e.g. temperature and time) were not considered. The lack of an optimization process remains problematic if the term "optimal" is used. This remark also stands for part given between lines 316-318. It is not clear in what manner the presented results lines with the following statement: ”...this study aims to determine the optimal processing methods to maximize the bioavailability..”. The Introduction is well written, with appropriate background and references related to the aim of presented work. Also, it is necessary to make a correction of the English language and grammar.
The following are same recommendations, that could, I believe, enhance the manuscript quality:
Line 151: Which water bath shaker was used?
Lines: 157-158: Explain what abbreviations P, L, M, W and S stand for.
Lines 160-188: The procedures for total phenols, total anthocyanins and total carotenoids determination should be given in more details.
Lines 189-192: The HPLC determination of phytochemicals must be explain in more details. The authors refer to their previous work, in which they adopted procedure from Espín et al. (Food Chem. 194 (2016), 1073-1080) who used HPLC-DAD-MS analysis.
Lines 194-200: This part also must be explained in more details. The given reference is inappropriate and should be changed. It is completely unclear what is EC50 and how it was assessed. The same remarks stand for determination of reducing power (lines 201-206). If EC50 is dose related than maybe paper https://doi.org/10.1002/bimj.201300123 can be useful.
Line 319: How the extraction yield was assessed?
Lines 321-324: The term “optimization” is used again. On the other hand, extraction solution when it comes about food, do not need to be just environmentally friendly but safe for human diet as well.
Line 340: Those are three groups of different compounds, not three different compounds.
Line 368: The 95% ethanol solution is not “optimal extraction condition”. Maybe appropriate term would be: “most suitable extractant”. This comment is in line with general observation regarding optimization.
Lines 376-391: Why only chlorogenic and rosmarinic acid were analyzed? If total phenols content was determined using Folin-Ciocalteu reagent, then it is better to use term total reducing power, since this reagent can react with substances other than phenols. But, since F-C method is widely used for assessing total phenol content (TP), then percentage of hydroxycinnamoyl derivatives in TP should be avoided. How recalculation based on extraction yield was performed, not only for phenols, but for anthocyanins and carotenoids as well?
Part 3.4: This part is not clear, since there is a lack of explanation how DPPH radical-scavenging ability and reducing power were assessed. It is not clear what was the aim of recalculations and what those recalculations means.
Line 472: The statement: “the effective concentration at which the absorbance is 0.5” is not supported with reference as well as it is not in line with previous explanation for EC50 (part 2.5)
Line 748: Only one extraction solvent (ethanol) at different concentrations was used for extraction of bioactive compounds. The authors should pay attention to this and change it through the whole manuscript.
Comments on the Quality of English Language
It is necessary to make a correction of the English language and grammar
Round 2
Reviewer 2 Report
Comments and Suggestions for Authors
The authors put an effort to improve manuscript and adequately address all comments. The manuscript is well written and can be accepted. The added and revised parts made this manuscript more comprehensive. The presented manuscript offers new insights into the antioxidant activity and enzyme inhibition properties of Tamarillo extracts.
Comments on the Quality of English Language
Minor corrections of English language and grammar.